# Antibody Response to COVID-19 mRNA Vaccines in Oncologic and Hematologic Patients Undergoing Chemotherapy

**Manlio Mencoboni** [1], **Vincenzo Fontana** [2], **Azzurra Damiani** [1], **Antonino Spitaleri** [3], **Alessandro Raso** [3], **Luigi Carlo Bottaro** [4], **Giovanni Rossi** [5], **Luciano Canobbio** [5], **Antonella La Camera** [1], **Rosa Angela Filiberti** [2,*], **Paola Taveggia** [1] and **Alessia Cavo** [1]

1   Oncology Unit, ASL 3, Villa Scassi Hospital, Corso Scassi 1, 16149 Genoa, Italy;
    manlio.mencoboni@fastwebnet.it (M.M.); azzurra.damiani@asl3.liguria.it (A.D.);
    antonella.lacamera@asl3.liguria.it (A.L.C.); paola.taveggia@asl3.liguria.it (P.T.);
    alessia.cavo@asl3.liguria.it (A.C.)
2   Clinical Epidemiology Unit, IRCCS Ospedale Policlinico San Martino, Largo Rosanna Benzi 12, 16100 Genoa,
    Italy; vincenzo.fontana@hsanmartino.it
3   Analysis Laboratory, ASL 3, Via Bertani 4, 16125 Genoa, Italy; antonino.spitaleri@asl3.liguria.it (A.S.);
    alessandro.raso@asl3.liguria.it (A.R.)
4   General Direction, Asl 3, Via Bertani 4, 16125 Genoa, Italy; direzione.generale@asl3.liguria.it
5   Oncology Unit, Antero Micone Hospital, Largo Nevio Rosso 2, 16100 Genoa, Italy;
    giovanni.rossi@asl3.liguria.it (G.R.); luciano.canobbio@asl3.liguria.it (L.C.)
*   Correspondence: rosangelafili@yahoo.it

**Abstract:** Background: Information on immune responses in cancer patients following mRNA COVID-19 vaccines is still insufficient, but generally, patients had impaired serological responses, especially those with hematological malignancies. We evaluated serological response to COVID-19 mRNA vaccine in cancer patients receiving chemotherapy compared with healthy controls. Methods: In total, 195 cancer patients and 400 randomly selected controls who had been administered a Pfizer-BioNTech or Moderna COVID-19 vaccines in two doses were compared. The threshold of positivity was 4.33 BAU/mL. Patients were receiving anticancer treatment after the first and second dose of the vaccines. Results: a TOTAL OF 169 patients (87%) had solid tumors and 26 hemolymphopoietic diseases. Seropositivity rate was lower in patients than controls (91% vs. 96%), with an age/gender-adjusted rate ratio (RR) of 0.95 (95% CL = 0.89–1.02). Positivity was found in 97% of solid cancers and in 50% of hemolymphopoietic tumors. Both advanced and adjuvant therapy seemed to slightly reduce seropositivity rates in patients when compared to controls (RR = 0.97, 95% CL = 0.89–1.06; RR = 0.94, 95% CL = 0.87–1.01). Conclusions: the response to vaccination is similar in patients affected by solid tumors to controls. On the contrary, hemolymphopietic patients show a much lower response than controls.

**Keywords:** COVID-19; SARS-COV-2; vaccine; cancer; anticancer treatment; seroconversion; antibodies

## 1. Introduction

Severe acute respiratory syndrome coronavirus 2 (SARS-CoV-2) is a positive sense, enveloped RNA beta coronavirus that emerged in Wuhan, China, in December 2019. It is the cause of the clinical disease known as COVID-19. It is plausible that the susceptibility of cancer patients to severe forms of COVID-19 is related more to a poor general status and comorbidities than to the cancer or therapies [1–4]. Nevertheless, the COVID-19 pandemic has a major impact at all stages of cancer treatment, and the impact on cancer healthcare systems, from screening programs to palliative therapies, is major [5].

Different therapies have been proposed to mitigate infective symptoms or post-viral sequelae of COVID-19 infection, including anticoagulants, anti-inflammatory drugs, antibodies contained in the plasma of convalescent patients, SARS-CoV-2-neutralizing monoclonal

antibodies, stem cell-based therapies, and biological drugs, but the different spectrum of disease severity does not permit to identify a unified therapeutic strategy [6,7]. There are only a few clinically approved drugs, and no treatment except corticosteroids and tocilizumab and the new oral antiviral agents (Molnupiravir, Paxlovid, and R emdesivir) has been shown with a high level of evidence to achieve a decreased rate of severe COVID-19 [8–12], so it is highly desirable that a potential vaccine can induce a potent antibody response as well as a long-term protection.

COVID-19 vaccines represent a major issue to decrease the risk of severe forms of the COVID-19 and to maintain a normal cancer care [13,14]. Nevertheless, there is concern that they are effective in preventing the infection in cancer patients.

High-risk patients with cancer who are candidates for priority access to vaccination are those treated by chemotherapy. The very high-priority population includes patients with curative treatment and palliative first- or second-line chemotherapy as well as patients requiring surgery or radiotherapy involving a large volume of lung, lymph node, and/or hematopoietic tissue [15].

However, cancer patients, like other vulnerable subpopulations, may respond in a different way to vaccines and are less likely to mount an optimal immune response necessary to confer immunity. This potential limited response must be balanced against the vulnerability of patients receiving chemotherapy and the risk of serious adverse outcomes from COVID-19.

Information on how novel mRNA vaccines elicit immune responses in cancer patients is still low, but generally, it has been shown that cancer patients had impaired serological responses respect to healthy individuals. In addition, the seroconversion rate was inferior in those with hematological versus solid cancers, particularly those following highly immunosuppressive therapies [16–24].

For this reason, it is necessary to implement cohorts with immunological and clinical monitoring of vaccinated cancer patients on active treatment.

In this observational study, we evaluated serological response to COVID-19 vaccination with mRNA vaccine in patients with solid and hematologic tumors receiving cancer chemotherapy in a real-world setting and compared it with the antibody production in healthy people who received same vaccines at the same time.

## 2. Materials and Methods

### 2.1. Study Groups

The study included a group of 195 cancer patients (cases) followed in two hospitals in Genova (Villa Scassi and Micone) and 400 subjects randomly selected from a group of 3736 healthcare workers (controls) who had been intramuscularly administered a Pfizer-BioNTech BNT162b2 or Moderna mRNA-273 SARS-CoV-2 vaccine in two doses given 3 weeks apart by 12 February 2021. A mass vaccination program against COVID-19 in healthcare workers has been introduced in the hospitals, while vaccines have been administered to eligible oncologic patients starting with those over 65 years of age. All participants completed the full vaccination.

Cases had a higher percentage of males (42.1% vs. 25.8%) (Table 1) and were older than controls (median/min-max = 71.8/35.5–93.7 years vs. 53.4/20.3–69.4 years).

The majority of cases had solid tumors (86.7%) with a greater prevalence of cancers of the breast (25.6%), colon-rectum (17.4%), lung (11.3%), and prostate (5.1%). Twenty-seven percent of patients had miscellaneous tumors: esophagus (2), stomach (5), pancreas (6), bladder (3), uterus (6, 2 leiomyosarcomas), ovary (9, 1 carcinosarcoma), testis (2), and mesothelioma (6). In addition, there was one case for each of the following tumor types: gastrointestinal stromal (GIST); papilla of vater, bilio-pancreatic, neuroendocrine, vulva, head, and neck; liposarcoma, sarcoma, Kaposi sarcoma, kidney, anus, thymus, melanoma, and of unknown primary origin.

**Table 1.** Characteristics of cancer patients and of healthcare workers.

| Characteristics | Cases (Cancer Patients) | | Controls (Healthcare Workers) | |
|---|---|---|---|---|
| | No | % | No | % |
| Age | | | | |
| 20–50 | 14 | 7.2 | 145 | 36.3 |
| 51–55 | 13 | 6.7 | 93 | 23.3 |
| 56–60 | 20 | 10.3 | 103 | 25.8 |
| 61–94 | 148 | 75.9 | 59 | 14.8 |
| Gender | | | | |
| Male | 82 | 42.1 | 103 | 25.8 |
| Female | 113 | 57.9 | 297 | 74.3 |
| Cancer site | | | | |
| HLP | 26 | 13.3 | - | - |
| Colon | 34 | 17.4 | - | - |
| Breast | 50 | 25.6 | - | - |
| Lung | 22 | 11.3 | - | - |
| Prostate | 10 | 5.1 | - | - |
| Other | 53 | 27.2 | - | - |
| Chemotherapy | | | | |
| Adjuvant | 54 | 27.7 | - | - |
| Advanced | 140 | 71.8 | - | - |
| Missing | 1 | 0.5 | - | - |
| Whole sample | 195 | 100.0 | 400 | 100.0 |

Legend: $n$/%, absolute and relative frequency; HLP, hemolymphopoietic.

Twenty-six cases had hemolymphopoietic (HLP) diseases, in particular: 3 Hodgkin's lymphomas (HL), 10 non-Hodgkin's lymphomas (NHL), 2 chronic lymphocytic leukemias (CLL), 3 myelomas, 2 essential thrombocythemia, and 6 of other histology, namely acute myeloid leukemia (AML), myelodysplasia, Waldenstrom macroglobulinemia (WM), monoclonal peak, plasmacytoma, and polycythemia vera (PV).

Adjuvant therapy was given to 27.7% of cases, advanced therapy to 71.8% (Table 1), and all were receiving chemotherapy of any kind at the time of vaccination.

Pfizer and Moderna vaccines were administered to 56% and 44% of subjects, respectively, in equal proportion between cases and controls. Cancer patients were receiving active anticancer treatment after the first and second dose of the vaccines.

### 2.2. Antibody Evaluation

Blood samples were taken on average 24.9 days (median = 21, SD = 10.3) after administration of the second dose.

Serum samples were analyzed with immunoassays CLIA (Snibe Diagnostic, Medical System), which use an automated platform Maglumi 800 for the detection of IgG against segment S-RBD (Receptor Binding Domain) of S1 protein, specific for SARS-CoV-2. Results are calculated referring to a calibration curve and expressed in BAU/mL (Binding Antibody Unit).

The threshold of positivity was 4.33 BAU/mL in accordance with the manufacturer's instructions. Samples higher than the limit of linearity (433 BAU/mL) were considered as highly positive.

The study was approved by the Ethical Board of National Infectious Diseases Spallanzani Institute, Rome, Italy, and was conducted in accordance with the Declaration of Helsinki and good clinical practice. Informed consent was obtained from all subjects involved in the study.

### 2.3. Statistical Analysis

Baseline characteristics of cancer patients (cases) and healthy subject (controls) were explored and summarized using descriptive statistics. In particular, all categorical variables

(e.g., gender, cancer site, vaccine type, etc.) were expressed in terms of absolute and relative frequencies (percentages). In addition, age at vaccination was considered as a four-category factor using 50, 55, and 60 years as cutoff points.

The distribution of IgG levels was described through geometric mean (GM) and corresponding 95% confidence limits (95% CL). Differences in IgG distribution in subgroups of subjects were assessed using the nonparametric Kruskal–Wallis test.

In order to evaluate the relationship between IgG levels and subjects' characteristics, antibody concentration was transformed into two binary serologic outcomes using 4.33 BAU/mL and 433.0 BAU/mL as cutoff values to define a positive or highly positive response, respectively, and a modified Poisson regression model was applied to both outcomes [25]. Next, seropositivity rate ratio (RR) was calculated as an index of association between antibody concentration and all explanatory variables (disease status/cancer site, age at vaccination, gender, and chemotherapeutic regimen) included in the Poisson model. A 95% CL was also computed for each RR and related statistical inference carried out using the likelihood ratio test [26].

All analyses were performed using Stata (StataCorp. Stata Statistical Software. Release 16.1. College Station, TX, USA, 2019).

## 3. Results

All cases were negative for prior COVID-19 infection.

Table 2 reports the distribution of IgG levels by age at vaccination, gender, disease status, cancer site, and chemotherapeutic treatment.

**Table 2.** Antibody levels after two doses of vaccine.

| Subjects' Characteristics | No. | % | GM (BAU/mL) | 95% CL (BAU/mL) |
|---|---|---|---|---|
| Age | | | | |
| 20–50 | 159 | 26.7 | 323.1 | 273.0–382.4 |
| 51–55 | 106 | 17.8 | 333.8 | 271.4–410.6 |
| 56–60 | 123 | 20.7 | 299.5 | 235.8–380.3 |
| 61–94 | 207 | 34.8 | 194.7 | 150.5–252.0 |
| Gender | | | | |
| Male | 185 | 31.1 | 139.5 | 189.5–302.7 |
| Female | 410 | 68.9 | 282.3 | 246.4–323.4 |
| Disease status | | | | |
| Controls | 400 | 67.2 | 319.3 | 283.6–359.4 |
| Cases | 195 | 32.8 | 187.6 | 144.3–243.9 |
| Cancer site | | | | |
| Controls | 400 | 67.2 | 319.3 | 283.6–359.4 |
| HLP | 26 | 4.4 | 12.7 | 4.4–37.2 |
| Colon | 34 | 5.7 | 311.6 | 202.7–479.2 |
| Breast | 50 | 8.4 | 274.1 | 186.6–402.3 |
| Lung | 22 | 3.7 | 257.7 | 177.9–373.2 |
| Prostate | 10 | 1.7 | 267.9 | 82.9–865.8 |
| Other | 53 | 8.9 | 290.6 | 199.6–423.1 |
| Chemotherapy | | | | |
| Controls | 400 | 67.2 | 319.3 | 283.6–359.4 |
| Adjuvant | 54 | 9.1 | 221.9 | 145.9–337.6 |
| Advanced | 140 | 23.5 | 174.8 | 125.7–243.0 |
| Whole sample | 595 | 100.0 | 268.2 | 238.3–302.0 |

Legend: GM, geometric mean; 95% CL, 95% confidence limits for GM; HLP, hemolymphopoietic.

Overall, patients had lower antibody response (GM, 95% CL = 187.6 BAU/mL, 144.6–243.9 vs. 319.3 BAU/mL, 283.6–359.4), and IgG concentration tended to halve with age, from 323.1 BAU/mL (95% CL = 273.0–382.4) for subjects < 51 years to 194.7 BAU/mL

(95% CL = 150.5–252.0) for those > 60 years. Additionally, a higher response rate was observed among females.

Patients with solid tumors had a GM concentration ranging from 257.7 BAU/mL (*n* = 22, 95% CL = 177.9–373.2) for lung cancer to 311.6 BAU/mL (*n* = 34, 95% CL = 202.7–479.2) for colorectal tumors. Lower concentration of antibodies was found for HLP malignancies (*n* = 26, GM = 12.7 BAU/mL, 95% CL = 4.4–37.2) and particularly for non-Hodgkin's lymphomas (*n* = 9, GM = 3.81 BAU/mL, 95% CL = 0.5–30.7). Among these tumors, the lowest values were found in single patients with monoclonal peak (0.82 BAU/mL), acute myeloid leukemia (1.16 BAU/mL), plasmacytoma (2.86 BAU/mL), and WM (2.94 BAU/mL).

Table 3 shows the results obtained through the Poisson regression analysis when applied to a positive (IgG > 4.33 BAU/mL) or highly positive (IgG > 433.0 BAU/mL) serologic response. Three different regression analyses were performed to appreciate differences in seroconversion: model 1 to compare cases with controls; model 2 to highlight the separate contribution of HLP patients and patients with other malignancies; and model 3 to evaluate the role of two different chemotherapeutic regimens (adjuvant or advanced) active during the vaccination period. In all models, age at vaccination, gender, and vaccination center were always considered.

**Table 3.** Joint effect of individual characteristics on a positive (IgG > 4.33 BAU/mL) or highly positive (IgG > 433.0 BAU/mL) serologic response estimated through the Poisson regression analysis.

| Model | Factor and Levels | Positive Response | | | | Highly Positive Response | | | |
|---|---|---|---|---|---|---|---|---|---|
| | | *n* | % | RR | 95% CL | *n* | % | RR | 95% CL |
| 1 | Age at vaccination | | | | | | | | |
| | 20–50 | 155 | 97.5 | 1.00 | (Ref.) | 128 | 80.5 | 1.00 | (Ref.) |
| | 51–55 | 103 | 97.2 | 0.99 | 0.96–1.04 | 93 | 87.7 | 1.10 | 0.99–1.22 |
| | 56–60 | 116 | 94.3 | 0.97 | 0.93–1.02 | 108 | 87.8 | 1.11 | 1.01–1.23 |
| | 61–94 | 188 | 90.8 | 0.95 | 0.89–1.00 | 144 | 69.6 | 0.99 | 0.87–1.13 |
| | Gender | | | | | | | | |
| | Male | 172 | 93.0 | 1.00 | (Ref.) | 142 | 76.8 | 1.00 | (Ref.) |
| | Female | 390 | 95.1 | 1.01 | 0.96–1.06 | 331 | 80.7 | 1.00 | 0.87–1.13 |
| | Disease status | | | | | | | | |
| | Control | 385 | 96.3 | 1.00 | (Ref.) | 345 | 86.3 | 1.00 | (Ref.) |
| | Case | 177 | 90.8 | 0.95 | 0.89–1.02 | 128 | 95.6 | 0.76 | 0.66–0.87 |
| 2 | Cancer site | | | | | | | | |
| | Control | 385 | 93.3 | 1.00 | (Ref.) | 345 | 86.3 | 1.00 | (Ref.) |
| | HLP | 13 | 50.0 | 0.53 | 0.36–0.78 | 5 | 19.3 | 0.23 | 0.10–0.51 |
| | Colon | 34 | 100.0 | 1.06 | 1.01–1.11 | 25 | 73.5 | 0.87 | 0.69–1.09 |
| | Breast | 48 | 96.0 | 1.00 | 0.93–1.08 | 34 | 68.0 | 0.80 | 0.65–0.99 |
| | Lung | 22 | 100.0 | 1.06 | 1.01–1.12 | 13 | 59.1 | 0.70 | 0.49–1.00 |
| | Prostate | 9 | 90.0 | 0.94 | 0.76–1.15 | 9 | 90.0 | 1.05 | 0.82–1.35 |
| | Other | 51 | 96.2 | 1.01 | 0.94–1.10 | 42 | 79.3 | 0.94 | 0.79–1.12 |
| 3 | Chemotherapy | | | | | | | | |
| | Control | 385 | 93.3 | 1.00 | (Ref.) | 345 | 86.3 | 1.00 | (Ref.) |
| | Adjuvant | 51 | 94.4 | 0.97 | 0.89–1.06 | 35 | 64.8 | 0.73 | 0.59–0.91 |
| | Advanced | 125 | 89.3 | 0.94 | 0.87–1.01 | 92 | 65.7 | 0.77 | 0.66–0.89 |

Legend: *n*/%, absolute/relative frequency of seropositive subjects; RR, seropositivity rate ratio adjusted for vaccination center; 95% CL, 95% confidence limits for RR; Ref., reference category; HLP, hemolymphopoietic. Note: RR of models 2 and 3 were also adjusted for age at vaccination and gender.

When assuming 4.33 as a cutoff value for seropositivity (Table 3), no remarkable difference in response was observed for gender (Model 1, female vs. male: RR = 1.01, 95% CL = 0.96–1.06), while an age-dependent decreasing trend was pointed out. In particular, a reduction of 5% (Model 1, RR = 0.95, 95% CL = 0.89–1.00) was estimated in the older subjects (>60 years) when compared to the younger subjects (<51 years). Very similar

results for gender and age were also derived from the other two regression models (data not shown).

As far as disease status is concerned (Table 3, Model 1), response rate was lower in cases than controls (90.8% vs. 96.3%), with an overall RR = 0.95 (95% CL = 0.89–1.02). In addition, positivity was found in at least 90% of solid cancers (from 90% for prostate cancer to 100% for colon and lung carcinomas), and only 50% of HLP tumors had an IgG level higher than the positivity threshold (Table 3, Model 2). With respect to the control seropositivity rate (96.3%), the latter figure amounted to a response rate reduction of about 50% (RR = 0.53, 95% CL = 0.36–0.78). By contrast, no substantial difference was found for all other cancer cases taken as a whole (RR = 1.02, 95% CL = 0.97–1.08) (data not shown).

Figure 1 shows variations in positive (IgG > 4.33 BAU/mL) or highly positive (IgG > 433 BAU/mL) serologic responses of each cancer group with respect the healthy controls.

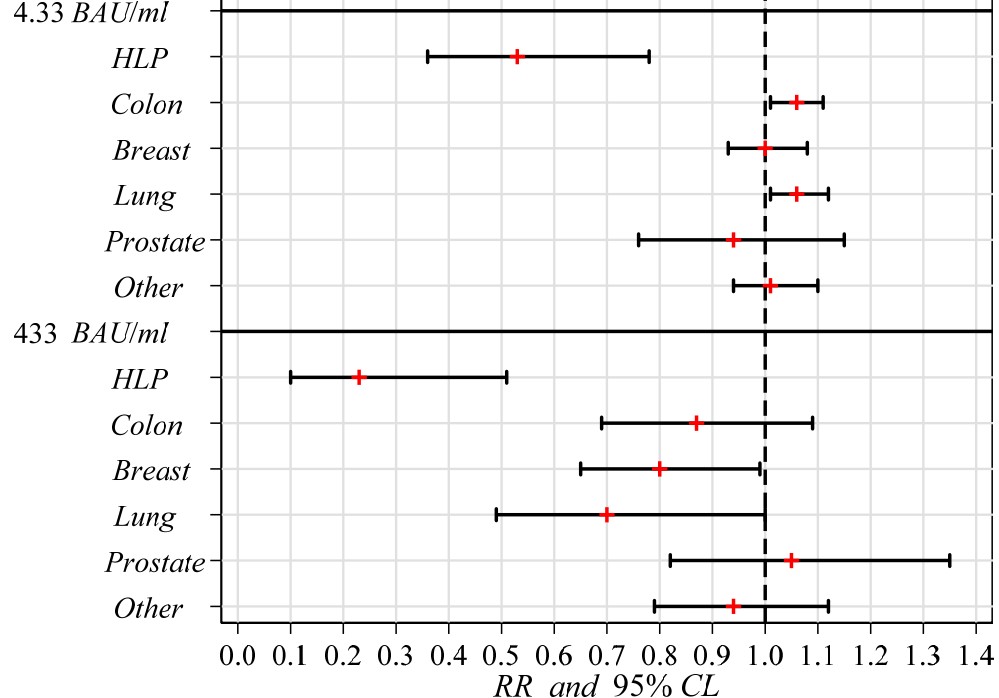

**Figure 1.** Caterpillar-plot of variations in positive (*IgG* > 4.33 *BAU/mL*) or highly positive (*IgG* > 433 *BAU/mL*) serologic responses of each cancer group with respect the healthy controls. Legend: *RR*, (red plus sign) ratio between the seropositivity rates of each cancer group and control group, estimated through the Poisson regression analysis; 95% *CL*, (whiskers) 95% confidence limits for RR; *RR* = 1, (vertical dashed line) rates in both groups are equal; *RR* < 1, rate in cancer group is less than that in control group; *RR* > 1, rate in cancer group is greater than that in control group. When the whiskers do not cross the vertical dashed line, the difference between the two rates is statistically significant.

Finally, patients receiving adjuvant therapy had a rate of seroconversion very similar to that of controls (RR = 0.97, 95% CL = 0.89–1.06), while advanced therapy seemed to induce a slightly more discernible reduction (RR = 0.94, 95% CL = 0.87–1.01) (Table 3, Model 3).

When an IgG titer of 433.0 was used as a threshold of seropositivity (highly positive response), no reduction in response was observed for age (RR = 0.99, 95% CL = 0.87–1.13 for subjects > 60 years). As already observed in the previous analysis, gender does not appear to play any noteworthy role (RR = 1.00, 95% CL = 0.87–1.13) (Table 3, Model 1).

In this context, cases experienced a seropositivity rate remarkably lower than controls (RR = 0.76, 95% CL = 0.66–0.87) (Table 3, Model 1), in particular for HLP tumors (RR = 0.23, 95% CL = 0.10–0.51), among which the frequency of responders was particularly low (19.3%) (Table 3, Model 2). Although in all other cancers, seropositivity rate was on average higher (72.8%) than that detected in HLP patients, an overall reduction of 15% (RR = 0.85, 95% CL = 0.74–0.97, data not shown) in comparison to controls was pointed out, especially for colon (73.5%, RR = 0.87, 95% CL = 0.69–1.09), breast (68.0%, RR = 0.80, 95% CL = 0.65–0.99), and lung (59.1%, RR = 0.70, 95% CL = 0.49–1.00) cancers (Table 3, Model 2).

Additionally, on this occasion, both chemotherapeutic regimens administrated at the time of vaccination seemed to play a role in attenuating seropositivity with similar response rates around 65%, which corresponded to a mean relative reduction in seroconversion of about 25% (adjuvant: RR = 0.73, 95% CL = 0.59–0.91; advanced: RR = 0.77, 95% CL = 0.66–0.89) (Table 3, Model 3) when compared to the control rate (86.3%).

## 4. Discussion

We evaluated antibody response after a second dose of the Pfizer–BioNTech BNT162b2 SARS-CoV-2 or Moderna mRNA COVID-19 vaccines in two groups subjects: 195 cancer cases and 400 healthcare workers (controls).

With a cutoff value for positive serologic finding of 4.33 BAU/mL, no differences were observed for gender, while response decreased with age. Positivity was found in 96% of controls, in at least 90% of solid cancers and in only 50% of patients with HLP tumors. Compared to controls, response was lower in patients undergoing advanced therapy. When considering a major cut-off of 433 BAU/mL, response was significantly lower both for carcinomas and for HLP tumors, with a positivity rate ranging from 59% to 90% for solid tumors and of 19% for HLP tumors.

Whether patients with a neoplastic disease are at higher risk of severe SARS-CoV-2 infection with inferior clinical outcomes and death is still uncertain [27]. Some studies have indicated that patients with cancer may experience a severe COVID-19 clinical course and higher death rates [28]. In particular, severe forms of COVID-19 were diagnosed in patients with metastatic tumors, lung cancer, or hematological malignancies and in patients receiving immunotherapy or chemotherapy or undergoing surgery [29]. Conversely, other authors reported that patients with cancer did not appear to have additional risk compared with the general population and suggested that the susceptibility of cancer patients to severe forms of COVID-19 are not due to the cancer per se or its treatments but to their poor general status and comorbidities. Indeed, almost all chronic health conditions are positively associated with an increased risk of COVID-19-related hospitalization and in-hospital mortality, and COVID-19 severity is predominantly associated with patients' age and cardiovascular comorbidities [2–4,30,31].

The outbreak of COVID-19 has led to the development of different types of vaccines, including mRNA vaccines produced by Pfizer–BioNTech (BNT162b2) and Moderna (mRNA-1273), which are the first mRNA vaccines authorized for use in the general public setting [32].

Two doses of the vaccine achieved a 94.8% and 94.1% efficacy, respectively, and were found to be safe, effective, and efficient in preventing COVID-19, including severe disease in the general population, as demonstrated in clinical trials [8,33–35].

Patients with neoplastic diseases are prioritized for vaccination [36], but so far, information on vaccine efficiency in these patients is limited since they were not represented in trials evaluating the efficacy and safety profile of SARS-CoV-2 vaccines.

Generally, antibody response to two doses of mRNA vaccine was found in patients with the most common solid tumor types also undergoing chemotherapy or in patients with B-cell malignancies distant from therapy or in remission even if the level of the humoral response may be lower than those for general population [16,37–39]. The fraction of patients with solid tumors undergoing different types of therapy, mainly with poorer performance

status, who fail to obtain seroconversion after SARSCoV-2 mRNA vaccines ranges from 6 to 30% [16,19–21,30,40].

Individuals with hematologic malignancies, especially patients with B-cell-depleting agents or those treated with cellular therapies, are at risk of developing severe COVID-19 and are less likely to show protective immune responses to vaccination than the general population because of disease-related or treatment-related immunosuppression [37–39,41]. A significantly lower response rate was found among patients with chronic lymphocytic leukemia respect to gender- and aged-matched healthy controls (52% vs. 100 after 2 doses of BNT162b2 vaccine). The response rate ranged from 79% in patients with clinical remission after treatment to 55% in treatment-naive patients to decline to 16% in patients under treatment at the time of vaccination, showing that disease activity and treatment influenced the response to the vaccine [42].

A recent meta-analysis (heterogeneity index = 55%) showed that the pooled response for hematological malignancies was 64% (95% CL = 59%–69%) vs. 96% (95% CL = 92%–97%) for solid cancer and 98% (95% CL = 96%–99%) for healthy controls. Outcome was different across hematological malignancies, with a response of 50% for chronic lymphocytic leukemia, of 58% to 61% for aggressive and indolent non-Hodgkin's lymphoma, of 76% for multiple myeloma, and reaching 83% for myeloproliferative neoplasms and 91% for Hodgkin's lymphoma [18].

In our series, response to vaccine was similar among male and females, but immunogenicity decreased after age 60 years. Contrary to our results, antibody titers were slightly higher in the female population of some studies [43,44].

Older age was associated with lower antibody titers in some papers [43,45], but age was not a confounder according to other studies [30,46,47].

Our results confirmed the previous findings on the role of COVID-19 vaccines in oncologic patients even when considering the differences in the assays used to define immunogenicity and the adoption of different thresholds to define seroconversion. Nevertheless, there are some limitations. The major remark is that we know that no patient had a prior COVID-19 infection, but we were unaware of serologic status of controls before the vaccine, so we cannot exclude the possibility that the higher seropositivity rate among controls may be due to a former COVID-19 exposure. Nevertheless, we think that results are still indicative of a lower seroconversion in oncologic patients with hematopoietic diseases. If controls had been positive for SARS-CoV-2 infection, the higher antibody titer could be attributable to the disease; nevertheless, antibodies titers in hematopoietic patients are lower also when they are compared with the other patients with solid tumor who had not have COVID-19 infection, with a 50% reduction in response when considering the cut-off of positivity of 4.33 BAU/mL.

Second, cancer patients were older and had a higher percentage of males than the cohort of healthcare workers population. Third, we do not have information on the health status and of possible use of drugs of controls.

## 5. Conclusions

As shown by some authors, results of the study showed that response to vaccination is similar in patients affected by solid tumors to controls. On the contrary, hemolymphopoietic patients show a much lower response than controls when both thresholds of positivity were used. In addition, we observed a decrease of response to vaccines with age but only with the cut-off of positivity of 4.33 BAU/mL. In this context, the antibody levels resulted to be about 5% lower in subjects older than 60 years when compared to younger ages.

Guidelines are in favor of vaccination for most of cancer patients [28]. It seems appropriate to vaccinate not only cancer patients with ongoing treatment or a treatment having been completed less than 3 years ago but also household and close contacts. When possible, SARS-CoV-2 vaccination should be carried out before cancer treatment begins or can be performed during chemotherapy while avoiding periods of neutropenia and lymphopenia. For organizational reasons, vaccination should be performed in cancer

care centers with messenger RNA vaccines. In general, live virus vaccines risk infecting patients and for this reason their administration is recommended only after 6 months from chemotherapy end date.

**Author Contributions:** Conceptualization, M.M., R.A.F., and V.F.; methodology, R.A.F. and V.F.; software, V.F., A.C., G.R., and A.S.; validation, M.M., R.A.F., and V.F.; formal analysis, R.A.F. and V.F.; investigation, A.C., L.C., A.D., G.R., and P.T.; resources, L.C.B.; data curation, A.L.C., A.S., and A.R.; writing—original draft preparation, R.A.F. and V.F.; writing—review and editing, M.M., R.A.F., and V.F.; visualization, V.F. and R.A.F.; supervision, M.M.; project administration, A.R. and A.S.; funding acquisition, L.C.B. All authors have read and agreed to the published version of the manuscript.

**Funding:** This research received no external funding.

**Institutional Review Board Statement:** The study was conducted in accordance with the Declaration of Helsinki and approved by the Institutional Review Board (or Ethics Committee) of Istituto Nazionale per le Malattie Infettive "Lazzaro Spallanzani" Rome. Date of approval: 24 February 2022, approval number n. 21_2022.

**Informed Consent Statement:** Informed consent was obtained from all subjects involved in the study. Written informed consent has been obtained from the patients to publish this paper.

**Data Availability Statement:** The data presented in this study are available on request from the corresponding author.

**Acknowledgments:** Antonio Messina and Elisabetta Ginocchio for editing.

**Conflicts of Interest:** The authors declare no conflict of interest.

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
