# Peer review of "Antibody Response to COVID-19 mRNA Vaccines in Oncologic and Hematologic Patients Undergoing Chemotherapy"

_curroncol, doi:10.3390/curroncol29050273_

Round 1

Reviewer 1 Report

Comments to the Author:

This study data are valuable and required. The main remark is that you indicated that controls possibly are not good choice, since it is not known about earlier sars-cov-2 infections in control study group. Additionally, some more references would be welcome.

Introduction:

There is probably more literature data than presented about antiviral therapies (Molnupiravir, Paxlovid, Fluvoxamine) for the severe COVID-19. The same objection stands for COVID-19 vaccines.

Material and methods:

The study group should be the first chapter name in this section.

Results:

The first part of the results includes patients’ characteristic and it should be part of the description of the study group, in Material chapter.

The sentence: “All cases were negative for prior COVID-19 infection.”… could be the beginning of the Results chapter.

Discussion:

In the end of this chapter is indicated that there are no data about earlier sars-cov-2 infections in control study group. It is a big problem and in this case, I found that the controls are not good or the conclusion is not well done.

Conclusion:

This conclusion isn’t derived from the current study and it should be change.

The question is: are the results of this study confirmed that there is too low response to covid-19 vaccines in patients older than 60 years with hemolymphopoietic tumors?.

Author Response

We thank the Reviewer for the suggestions. In the following we will address the reviewer specific comments

This study data are valuable and required. The main remark is that you indicated that controls possibly are not good choice, since it is not known about earlier sars-cov-2 infections in control study group.

Additionally, some more references would be welcome.

Introduction:

1, There is probably more literature data than presented about antiviral therapies (Molnupiravir, Paxlovid, Fluvoxamine) for the severe COVID-19. The same objection stands for COVID-19 vaccines.

Answer: we have modified the sentences and added references as follows:

 Therapy: Different therapies have been proposed to mitigate infective symptoms or post-viral sequelae of COVID-19 infection, including anticoagulants, anti-infiammatory drugs, antibodies contained in the plasma of convalescent patients, SARS-CoV-2-neutralising monoclonal antbodies, stem cell-based therapies, biological drugs, but the different spectrum of disease severity does not permit to identify a unified therapeutic strategy [6,7]. There are only a few clinically approved drugs and no treatment, except corticosteroids and tocilizumab and the new oral antiviral agents (Molnupiravir, Paxlovid, and remdesivir), has been shown with a high level of evidence to achieve a decreased rate of severe COVID-19 [8-12], so it is highly desirable that a potential vaccine can induce a potent antibody response as well as a long-term protection.

 Alfano, G., Morisi, N., Frisina, M., Ferrari, A., Fontan, F., Tonelli, R., Franceschini, E., Meschiari, M., Donati, G., Guaraldi, G. Awaiting a cure for COVID-19: therapeutic approach in patients with different severity levels of COVID-19. Infez Med. 2022, 30, 11-21. doi: 10.53854/liim-3001-2.

 Drożdżal, S., Rosik, J., Lechowicz, K., Machaj, F., Szostak, B., Przybyciński, J., Lorzadeh, S., Kotfis, K-, Ghavami, S., Łos, M.J. An update on drugs with therapeutic potential for SARS-CoV-2 (COVID-19) treatment. Drug Resist Updat. 2021, 59, 100794. doi: 10.1016/j.drup.2021.100794.

Moreno, S., Alcáza, B., Dueñas, C., González Del Castillo, J,, Olalla, J,, Antela, A. Use of Antivirals in SARS-CoV-2 Infection. Critical Review of the Role of Remdesivir. Drug Des Devel Ther. 2022, 16, 827-841. doi: 10.2147/DDDT.S356951.

 Deng, J., Heybati, K., Ramaraju, H.B,, Zhou, F., Rayner, D., Heybati ,S. Differential efficacy and safety of anti-SARS-CoV-2 antibody therapies for the management of COVID-19: a systematic review and network meta-analysis. Infection. 2022, 19, 1–15. doi: 10.1007/s15010-022-01825-8.

 Robinson, P.C, Liew, D.F.L., Tanner, H.L., Grainger, J.R., Dwek, R.A., Reisler, R.B., Steinman, L., Feldmann, M., Ho, L.P., Hussell, T., Moss, P., Richards, D., Zitzmann, N. COVID-19 therapeutics: Challenges and directions for the future. Proc Natl Acad Sci U S A. 2022, 119, e2119893119. doi: 10.1073/pnas.2119893119.

 Zhou, H., Ni,W.J., Huang, W., Wang, Z., Cai, M., Sun, Y.C. Advances in Pathogenesis, Progression, Potential Targets and Targeted Therapeutic Strategies in SARS-CoV-2-Induced COVID-19. Front Immunol. 2022, 13, 834942. doi:10.3389/fimmu.2022.834942.

 Wen Wen Chen Chen Jiake Tang Chunyi Wang Mengyun Zhou 3Yongran Cheng Xiang ZhouQi WuXingwei Zhang Zhanhui FengMingwei WangQin Mao Efficacy and safety of three new oral antiviral treatment (molnupiravir, fluvoxamine and Paxlovid) for COVID-19; a meta-analysis. Ann Med. 2022, 54, 516-552. doi: 10.1080/07853890.2022.2034936.

 Vaccine: Information on how novel mRNA vaccines elicit immune responses in cancer patients is  still low, but generally it has been shown that cancer patients had impaired serological responses respect to healthy individuals. In addition. the seroconversion rate was inferior in those with haematological versus solid cancers, particularly those following highly immunosuppressive therapies [16-24].

 Mair, M.J., Berger, J.M., Berghoff, A.S., Starzer, A.M., Ortmayr, G., Puhr, H.C., Steindl, A., Perkmann, T., Haslacher, H., Strassl, R., Tobudic, S., Lamm, W.W., Raderer, M., Mitterer, M., Fuereder, T., Fong, D., Preusser, M. Humoral Immune Response in Hematooncological Patients and Health Care Workers Who Received SARS-CoV-2 Vaccinations. JAMA Oncol. 2022,  8, 106-113. doi: 10.1001/jamaoncol.2021.5437.

 Thakkar, A., Gonzalez-Lugo, J.D., Goradia, N., Gali, R, Shapiro L..C, Pradhan, K., Rahman, S., Kim, S.Y., Ko, B., Sica, R.A., Kornblum, N., Bachier-Rodriguez, L., McCort, M., Goel, S., Perez-Soler, R., Packer, S., Sparano, J., Gartrell, B., Makower, D., Goldstein, Y.D., Wolgast, L., Verma, A., Halmos, B. Seroconversion rates following COVID-19 vaccination among patients with cancer. Cancer Cell. 2021, 39, 1081-1090.e2. doi: 10.1016/j.ccell.2021.06.002.

 Seneviratne, S.L., Yasawardene, P., Wijerathne, W., Somawardana,B. COVID-19 vaccination in cancer patients: a narrative review. J Int Med Res. 2022, 50, 3000605221086155. doi: 10.1177/03000605221086155.

 2. Material and methods:

The study group should be the first chapter name in this section.

Answer: we have changed as suggested

 Results:

  1. The first part of the results includes patients’ characteristic and it should be part of the description of the study group, in Material chapter.

Answer: we have described case and controls in Materials and methods (2.1 Study groups)

 4. The sentence: “All cases were negative for prior COVID-19 infection.”… could be the beginning of the Results chapter.

Answer: we have moved the sentence in results

 Discussion:

  1. In the end of this chapter is indicated that there are no data about earlier sars-cov-2 infections in control study group. It is a big problem and in this case, I found that the controls are not good or the conclusion is not well done.

Answer: as already said in discussion, we agree with this remark, nevertheless, we think that results are indicative of a lower seroconversion in oncologic patients, in particular those with hematopoietic diseases. If controls had been positive for sars-cov-2 infections, the higher antibody titer could be attributable to the infection, nevertheless antibodies titers in hematopoietic are different also when they are compared with the other patients with solid tumor who had not have Covid infection, with a 50% reduction in response when considering the cut-off of positivity of 4.33 BAU/ml.

 In Discussion we wrote: "The major remark is that we know that no patient had a prior COVID-19 infection, but we were unaware of serologic status of controls before the vaccine, so we cannot exclude the possibility that the higher seropositivity rate among controls may be due to a former COVID-19 exposure. Nevertheless, we think that results are still indicative of a lower seroconversion in oncologic patients with hematopoietic diseases. If controls had been positive for SARS-CoV-2 infection, the higher antibody titer could be attributable to the disease, nevertheless antibodies titers in hematopoietic patients are lower also when they are compared with the other patients with solid tumor who had not have COVID-19 infection, with a 50% reduction in response when considering the cut-off of positivity of 4.33 BAU/ml.

 Conclusion:

  1. This conclusion isn’t derived from the current study and it should be change.

The question is: are the results of this study confirmed that there is too low response to covid-19 vaccines in patients older than 60 years with hemolymphopoietic tumors?

Answer: conclusion has been changed as follows: “As shown by some authors, results of the study showed that response to vaccination is similar in patients affected by solid tumors to controls. On the contrary, hemolymphopoietic patients show a much lower response than controls when both thresholds of positivity were used. In addition, we observed a decrease of response to vaccines with age, but only with the cut-off of positivity of 4.33 BAU/ml. In this context, the antibody levels resulted to be about 5% lower in subjects older than 60 years when compared to younger ages

Guidelines are in favor of vaccination for most of cancer patients [28]. It seems appropriate to vaccinate not only cancer patients with ongoing treatment or a treatment having been completed less than 3 years ago, but also household and close contacts. When possible, SARS-CoV-2 vaccination should be carried out before cancer treatment begins or can be performed during chemotherapy while avoiding periods of neutropenia and lymphopenia. For organizational reasons, vaccination should be performed in cancer care centers with messenger RNA vaccines. In general, live virus vaccines risk infecting patients and for this reason their administration is recommended only after 6 months from chemotherapy end date.”

Reviewer 2 Report

This manuscript mainly focuses on antibody response to Covid-19 mRNA vaccines in oncologic and hematologic patients undergoing chemotherapy. This project is interesting and some improvements need to be done before published on Current Oncology.

  1. The authors got the quantitation data with CLIA, it’s better to provide the data and have a deep analysis.

    2. Some units are missing in the manuscript. 

    3. It's better to organize the data into graphs to better clarify the conclusion and help readers get the information clearly.

    4. What's the difference of antibody response between patients with different kinds of solid tumors?

    5. How the authors choose the cut of values (4.33 and 433)?

Author Response

We thank the Reviewer for the suggestions. In the following we will address the reviewer specific comments

Comments and Suggestions for Authors

This manuscript mainly focuses on antibody response to Covid-19 mRNA vaccines in oncologic and hematologic patients undergoing chemotherapy. This project is interesting and some improvements need to be done before published on Current Oncology.

1) The authors got the quantitation data with CLIA, it’s better to provide the data and have a deep analysis.

Answer: we are not sure to have understood the question. Statistical analysis has been done after a binary transformation of original data (below and above a cut-off) since we wanted to verify the frequency of subjects among cases and controls exceeding the thresholds of 4.33 BAU/ml  (positivity) and of 433 BAU/ml (value corresponding to  the limit of linearity, we defined as highly positive). These values had been defined by the manufacturer. We think that other analyses on truncated/censored measurement data (i.e. Tobit analysis) were less appropriate for our aim.

 2) Some units are missing in the manuscript

Answer: missing units have been added

3) It's better to organize the data into graphs to better clarify the conclusion and help readers get the information clearly.

Answer: a caterpillar-plot of variations in positive (IgG > 4.33 BAU/ml) or highly positive (IgG > 433 BAU/ml) serologic responses of each cancer group with respect the control group has been added in results as Figure 1.  We wrote: “Figure 1 shows variations in positive (IgG > 4.33 BAU/ml) or highly positive (IgG > 433 BAU/ml) serologic responses of each cancer group with respect the healthy controls”.

4) What's the difference of antibody response between patients with different kinds of solid tumors?

Answer: as reported on Table 3 and described in results, when IgG > 4.33 BAU/ml was used as a cut-off point, positivity rate ranged from 90% for prostate cancer to 100% for colon and lung carcinomas, but only 50% of HLP patients showed a seroconversion. In addition, the seroconversion rate decreased to 19% when IgG threshold was 433 BAU/ml.

 5. How the authors choose the cut of values (4.33 and 433)?

Answer: antibody titers higher than 4.33 BAU/ml were  considered positive in accordance with the manufacturer’s instructions based on 95% clinical sensitivity and  specificity obtained testing 351 true positive and 229 true negative samples. Values higher than 433 BAU/ml were considered highly positive being samples exceeding the limit of linearity of the test.

Reviewer 3 Report

In my opinion, the paper entitled "Antibody response to Covid 19 mRNA vaccines in oncologic 2 and hematologic patients undergoing chemotherapy" is able to be published in the present form. Although there is extensive research on Covid 19 mRNA vaccines, this paper deals the evaluation of serological response to COVID-19 16 mRNA vaccine in cancer patients receiving chemotherapy compared against healthy controls in a detailed and comprehensive way. I especially appreciate the clarity of the presentation of the methodology, as well as the results and discussions. Moreover, the conclusions are strongly supported by the results obtained. 

Author Response

In my opinion, the paper entitled "Antibody response to Covid 19 mRNA vaccines in oncologic 2 and hematologic patients undergoing chemotherapy" is able to be published in the present form. Although there is extensive research on Covid 19 mRNA vaccines, this paper deals the evaluation of serological response to COVID-19 16 mRNA vaccine in cancer patients receiving chemotherapy compared against healthy controls in a detailed and comprehensive way. I especially appreciate the clarity of the presentation of the methodology, as well as the results and discussions. Moreover, the conclusions are strongly supported by the results obtained. 

We thank the reviewer for the comments

Round 2

Reviewer 1 Report

The authors considered all comments and suggestions and corrected the manuscript according to the instructions. I suggest that the manuscript be published in this form.

Reviewer 2 Report

The manuscript can be accepted.